# The Correlation between Oral Self-Harm and Ethnicity in Institutionalized Children

**DOI:** 10.3390/children8010002

**Published:** 2020-12-23

**Authors:** Alexandra Mihaela Stoica, Oana Elena Stoica, Ramona Elena Vlad, Anca Maria Pop, Monica Monea

**Affiliations:** 1Department of Odontology and Oral Pathology, George Emil Palade University of Medicine, Pharmacy, Science, and Technology of Târgu Mureș, 540139 Târgu Mureș, Romania; alexandra.stoica@umfst.ro (A.M.S.); ramona.vlad@umfst.ro (R.E.V.); monica.monea@umfst.ro (M.M.); 2Department of Pedodontics, George Emil Palade University of Medicine, Pharmacy, Science, and Technology of Târgu Mureș, 540139 Târgu Mureș, Romania; oana.stoica@umfst.ro; 3Faculty of Medicine, George Emil Palade University of Medicine, Pharmacy, Science, and Technology of Târgu Mureș, 540139 Târgu Mureș, Romania

**Keywords:** self-injurious behavior, institutionalized child, oral manifestations

## Abstract

Oral self-harm was described in institutionalized children who share a lack of emotional attention; frequently these children experience feelings such as neglect, loneliness, isolation or lack of connection with the world. The aim of our paper was to conduct a cross-sectional study in order to assess the prevalence of this behavior and its correlation with ethnicity among children from three institutions located in the central part of Romania. We examined 116 children from three ethnic groups, Romanians, Hungarians and local Roma population aged between 10–14 years old. The oral soft tissues were evaluated by one dentist who recorded the lesions of lips, buccal mucosa, commissures and tongue; data were statistically analyzed at a level of significance of *p* < 0.05. We found oral self-harm lesions in 18.1% participants, with statistically significant higher odds in girls (*p* = 0.03). The results showed an association between ethnicity and the development of these lesions (Chi-square *p* = 0.04). The most frequent lesions were located at oral commissures (35.48%), buccal mucosa (29.03%) and upper lip (19.36%). Oral self-harm lesions have a high incidence among institutionalized children in Romania. Identification of these cases in early stages is important, as these conditions are known to be aggravated during adolescence and adulthood.

## 1. Introduction

Self-mutilation is defined as a behavioral disturbance that consists of self-induced damage to body tissues, which might be associated in some cases with a conscious intent to commit suicide. Also called self-harm, it includes any intentional injury to one’s own body [1,2,3,4]. Historically, the first institutions for abandoned children can be traced back in Europe since the Middle Ages; they came into prominence in the 19th century in Western Europe, today being common in different parts of the world such as Asia, Central and South America, the Middle East or Africa. In the United States, orphanages were documented in the first half of the 20th century [5].

At present, worldwide there are between 8–10 million children living in different types of institution [6] and there is much scientific evidence that their psychological development is impaired by these life conditions [7]; furthermore, the trend of placing children in institutions appears to be growing [8,9]. According to data from literature, institutional care in Romania was associated with an impairment of the physical development [10] and also children who spent more than 6 months in an institution had higher rates of autism symptoms, inattention or disinhibited social engagement [11]. Due to the demands of taking care of a large number of children, the caregivers rarely interact with children in a warm manner, as their activity is frequently limited to routine care, such as feeding or toileting [12]. Therefore, most institutionalized children experience poor caregiver-child interaction and their physical, cognitive and social development is often delayed. Moreover, scientific data showed that these results are caused mainly by the quality of caregiver-child relationships, rather than by the quality of medical care and nutrition [13]. The inability to live with their parents predisposes institutionalized children to low self-esteem and impaired psychosocial development (attention problems or lower intelligence quotient) [7], which might represent confounding factors in the analysis of the correlation between self-harm and institutionalization.

In institutionalized children, the relief of emotional pain could be expressed by self-harm, as the physical wounds they create on themselves is a sign of their emotional suffering [14]. The self-harm behavior has many causes, including stressful life events or mental disorders such as depression or anxiety [15]. Adolescents use deliberate self-harm methods such as cutting, poisoning or overdosing, while children usually scratch or bite themselves; this phenomenon may start during childhood and intensifies in adolescents and young adults, girls being considered more vulnerable to this behavior than boys [16]. Among the etiological factors of deliberate self-harm the following conditions were included: depression, low self-esteem and sense of persistent hopelessness, attempts to seek help from others, poverty, abuse, attempts to resist suicidal thoughts and family dysfunction. The early detection of non-suicidal self-injury (NSSI) allows immediate intervention which might help these children stop this behavior. Left undiagnosed for a long period of time, NSSI becomes more frequent, severe and versatile, with negative consequences on the quality of life and more difficult recovery [17].

Oral self-harm (OSH) in institutionalized children occurs in connection with emotional, behavioral or even organic disorders. To date, most of the information comes from case series presentations and there are little scientific data regarding the frequency of OSH among abandoned children without mental disorders or retardation. Therefore, the aim of our paper was to conduct a cross-sectional study in order to assess the frequency and type of OSH among institutionalized children from three Romanian state centers. The null hypothesis to be tested was that there is no statistically significant difference regarding the prevalence of oral self-inflicted lesions, according to gender and ethnicity in children at puberty.

## 2. Materials and Methods

### 2.1. Study Design and Participants

Our investigation was conducted between December 2019–February 2020 in the Clinic of Odontology and Oral Pathology from the George Emil Palade University of Medicine, Pharmacy, Science, and Technology of Târgu Mureș, where there is a special program dedicated to dental medical care for institutionalized children, belonging to three state centers. The investigation was carried out after the approval obtained from the Ethics Committee of our university (No. 520/21.11.2019), accompanied by a written consent for the use of personal data signed in each case by the legal representative of the child (institution manager or legal guardian). Prior to enrolment in the study, children were also asked if they agreed to participate. We are located in the historical province of Transylvania, characterized by a multicultural and multiethnic population, represented mainly by Romanians, Hungarians and regional Roma. In order to address a source of bias related to the number of participants from each ethnicity, we decided to include close numbers in each group, according to age and gender. Moreover, all clinical examinations were carried out by one experienced dentist and data were recorded by one dental specialist. In our study we included 116 children aged between 10–14 years old, selected from a total of 167 children, based on application of inclusion criteria (status of institutionalized child for more than 5 years, age 10–14 years) and exclusion criteria (history of psychological counseling or psychiatric treatment, recordings of drugs or alcohol abuse, children with diagnosed neurologic or psychiatric disorders, known to be etiological factors of self-harm behavior, such as epilepsy, depression, anxiety or autism spectrum disorder) (Figure 1).

### 2.2. Clinical and Histopathological Examination

Ordinary dental examinations, with an emphasis on the health status of the lips, buccal mucosa and tongue (ulcerations, color change, surface aspect), were performed. In order to detect any changes from normal texture, the area between oral commissures was carefully evaluated by palpation. Cases in which a chronic evolution was suspected, resembling premalignant lesions, were further investigated by exfoliative cytology, using Papanicolau stain. All children who presented OSH were further referred to interdisciplinary evaluation by a psychologist and dental specialist.

### 2.3. Statistical Analysis

Statistical analysis was carried out using GraphPad Prism 7 for Windows (GraphPad Software, San Diego, CA, USA), by Fisher’s exact test and Chi-square test. The continuous variables were expressed as mean ± standard deviation and categorical variables as percentages and frequency distribution. The level of statistical significance was set at a *p* value < 0.05 (two-tailed).

## 3. Results

The distribution of the study group based on gender and ethnicity is presented in Table 1.

The presence of OSH was noticed in 21 participants (18.1%). According to Fisher’s exact test, girls had statistically significant higher odds of presenting OSH than boys (odds ratio (OR) = 3.268, 95% confidence interval (CI): 1.108–9.643, *p* = 0.03) (Table 2).

The results of our study showed that the presence of self-inflicted oral lesions is influenced by ethnicity (Chi-square original *p* = 0.04). After applying the Bonferroni correction, the level of statistical significance was adjusted at *p* < 0.0167. Therefore, Romanians had statistically significant lower odds of developing oral self-injuries compared to Roma participants (OR = 0.15, 95% CI: 0.03–0.75, *p* = 0.0164), but there was no significant difference neither between Romanians and Hungarians (OR = 0.22, 95% CI: 0.04–1.148, *p* = 0.07), nor between Hungarians and Roma participants (OR = 0.71, 95% CI: 0.24–2.06, *p* = 0.6) (Table 3).

In Table 4 the types of encountered lesions are summarized, based on location and frequency.

Most lesions were observed at the level of oral commissures (35.48%), followed by buccal mucosa (29.03%) and the upper lip (19.36%). The lowest value was obtained for the frequency of tongue lesions (3.23%). Suggestive clinical and histopathological aspects are presented in Figure 2 and Figure 3a,b.

## 4. Discussion

In Romania there is a large number of institutionalized children and a lack of scientific information regarding the consequences of this policy on oral health. Moreover, central Romania is multicultural and different minorities among which Hungarians and local Roma population are the most numerous. This allows a better assessment of more variables, in the effort to find possible risk factors for the development of OSH. The last few decades have been marked by increased scientific information on self-harm behavior, which could be the result of the interest of specialists or better diagnostic methods. Although considered pathological, it was reported that a large number of individuals have experienced a self-harm behavior at least once or even for a period of time in their life [18,19]. To be considered self-injury, a lesion must have the following characteristics: repetitive, socially unacceptable and to cause mild/moderate tissue damage [20]. Therefore, these lesions are usually hidden, the exact prevalence in the world is unknown and is believed to be underestimated [21]. Recent studies reported different percentages depending on the group of population analyzed, ranging from 4% in adults, 17–38% in students, 7.7–22.8% in institutionalized patients with mental disorders to 69% in high-risk young people (victims of sexual abuse or drug users) [22,23,24,25]. NSSI is a relatively common and insidious pervasive, often concealed habit that may start in childhood and increase in adolescence and young adulthood. Adolescent girls seem more vulnerable and the key components of NSSI behavior are represented by negative emotion and saturnine self-derogation [26,27].

According to data from literature, ethnicity might have an influence on self-harm behavior [28]. This was confirmed by the results of our study, as the group of local Roma showed statistically significant higher odds of developing OSH lesions compared to Romanians. The influence of ethnicity upon self-injury behavior was further confirmed by Toth et al. [29] who found that Roma population from Hungary is characterized by higher odds of developing suicidal behavior compared to non-Roma ethnics. The authors mention that studies from the UK and Hungary partially explain these tendencies by the high incidence of anxiety, depression and hostility from the majority population. For the Roma population in particular, the family concept has an important social value and, therefore, the lack of cohesion with relatives experienced by institutionalized children could be a strong negative factor for the development of anxiety and depression. These problems aggravate during adolescence and adulthood as a result of poverty, low educational level, and unemployment.

In a meta-analysis, published by Lang and Yao in 2018 [30], the estimated prevalence of NSSI in Chinese middle-school students was 22.37%, considered relatively high, females being more susceptible to this behavior (21.9% compared to a prevalence of 20.6% reported in male students). The results of our study are in accordance with this data, as out of the OSH overall prevalence of 18.1%, 13.79% were attributed to female participants and only 4.31% to male participants.

OSH is not a frequently encountered phenomenon in the daily clinical practice, but it can represent the first manifestation of a psychiatric disorder. AlSadhan et al. [31] found a higher prevalence of OSH among institutionalized children from Saudi Arabia, including gingival or mucosal lesions, cheek and lip biting. Traumatic lesions of the lips, accompanied by loss of tissue were recognized by many authors as the most frequent injuries of the oral mucosa. [32] This was explained by the proximity of incisors and canines, teeth with sharp cusps and incisal margins. In our study, the distribution of injuries is in accordance with scientific data, as the oral commissure and lips were affected in 35.48% and 32.26% cases, respectively, while the tongue was injured only in 3.23% cases. The frequent lesion of the oral commissure could be explained by the presence of caliculus angularis, a small projection of keratinized mucosa, easily injured between upper and lower canine, associated with a decreased level of pain.

Based on the literature, children who self-harm claim to have little to no pain while they are hurting themselves but they feel tension and anger towards themselves or others. This was observed also by our investigators, as none of the children who presented with OSH complained about pain during examination. A drawback of our study is that during the oral examination no psychological assessment was performed and, therefore, the tension or anger could not be quantified. It is estimated that the incidence of habitual self-injurers is nearly 1% of the population with a higher proportion of females than males, the typical onset of self-harming acts is usually at puberty. This behavior lasts 5–10 years but it can persist much longer without the proper treatment [33,34]. Institutionalized children show an increased prevalence of oral habits and OSH, which indicate emotional stress. Moreover, foster caregivers frequently lack information on these subjects and are unable to provide the proper support for these children [31].

In a study from 2015, Tortorici et al. [35] reported that oral soft tissue injuries had an incidence estimated at 2.5% in the Caucasian population. In our study group, a chronic evolution was suspected in 6 cases with OSH (5.17%) and these were further investigated using exfoliative cytology, the results confirming the benign evolution, with mild inflammation. The bite of the lips and buccal mucosa can destroy the superficial epithelium and if this parafunction has a chronic evolution, it can cause keratinized shreds or erosive and desquamative areas. These lesions can be easily identified by clinical inspection and are often related to psychologically tense persons [36]. However, in some cases, the lesions were mistaken for serious medical conditions and biopsies were required in order to rule out a malignancy. Therefore, it is important to perform a thorough clinical examination and to interpret the laboratory tests clearly [37].

### 4.1. Clinical Relevance

Identification of NSSI in the early stages is of utmost importance as scientific data confirm that up to 70% of these persons experienced also suicidal attempts [33]. Data from literature suggests that early age at which children engage in NSSI represents a risk factor for more episodes of NSSI during the lifetime with increased severity [17]. As self-injuries are conducted mainly in secret and may not be clearly visible, the periodic oral examination of these children might be useful in early diagnosis. Although minor OSH does not lead to a serious loss of tissue, it may affect oral health in the long run, with important social and emotional implications. Our results raise the question regarding efficient preventive measures, such as better training of caregivers and policies focusing on the psychosocial well-being of institutionalized children. Measures aiming to enhance subjective happiness and satisfaction with life at any age might decrease the prevalence of NSSI among institutionalized children [17].

### 4.2. Strengths and Limitations

According to our knowledge there were no studies addressing self-harm behavior of oral soft tissues in children from central Romania. However, our study encountered several limitations: the small sample size and cross-sectional design, which allowed a rapid and cost-effective evaluation of the prevalence of OSH, but is unable to provide a clear association between investigated variables. Therefore, longitudinal studies on the general population are required for a better understanding of these emotional disorders. Moreover, cognitive assessment of these children would have been useful in order to adjust confounding factors.

## Figures and Tables

**Figure 1 children-08-00002-f001:**
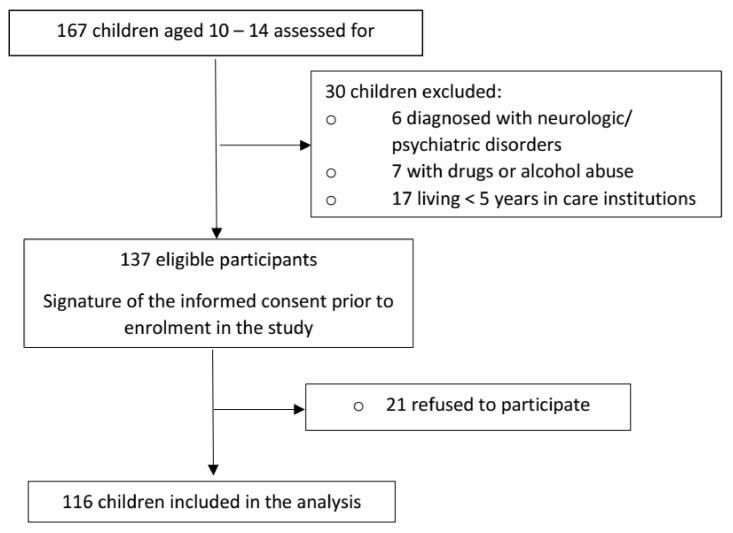
Flow diagram illustrating selection of participants to the study.

**Figure 2 children-08-00002-f002:**
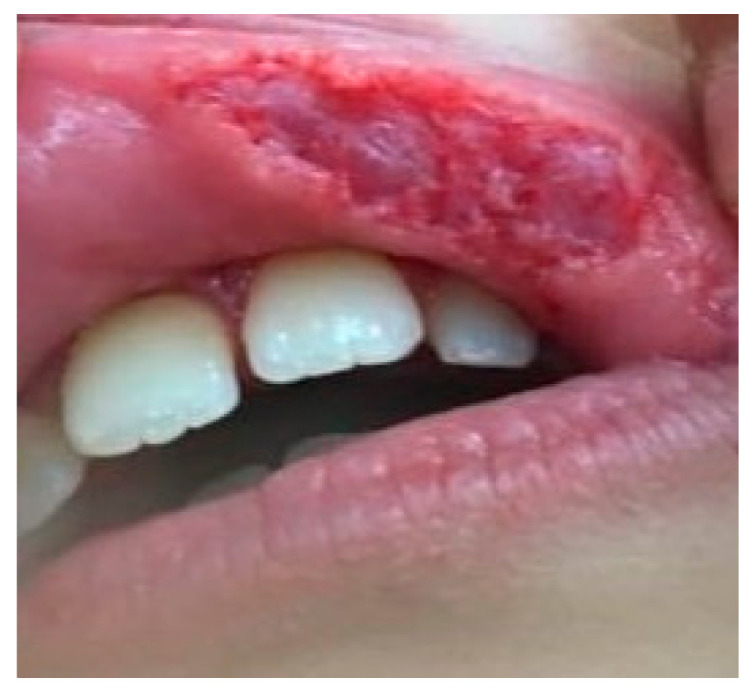
14-year-old female patient, with a large wound on her upper lip, non-bleeding area of 1 × 2 cm on the non-keratinized mucosa, surrounded by erythema that did not involve the vermilion border.

**Figure 3 children-08-00002-f003:**
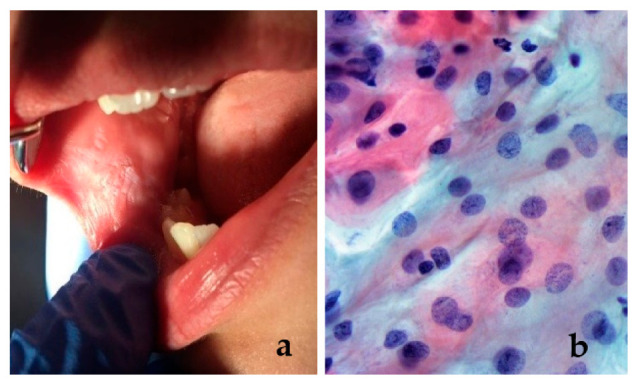
(**a**) A 12-year-old female patient with a white “patch” on the buccal mucosa, resembling leukoplakia; histopathological examination was performed in order to obtain the correct diagnosis; (**b**) exfoliative cytology did not confirm the presence of a premalignant lesion. Intermediate squamous cells (reflecting the accelerate turnover) with slight inflammation (different shape of nuclei, stainability, irregular contour of the nuclear border) (Papanicolaou stain, ×40).

**Table 1 children-08-00002-t001:** Distribution of the study group according to gender and ethnicity.

Gender/Ethnicity	Romanians	Hungarians	Roma	Total
Female	20 (17.24%)	19 (16.38%)	24 (20.69%)	63 (54.31%)
Male	17 (14.65%)	15 (12.93%)	21 (18.11%)	53 (45.69%)
Total	37 (31.89%)	34 (29.31%)	45 (38.8%)	116 (100%)

The mean age of the study group was 12.11 ± 1.36 years.

**Table 2 children-08-00002-t002:** The distribution of lesions according to gender.

Gender/Oral Lesion	Oral Lesion Present	Oral Lesion Absent	Total
Female	16 (13.79%)	47 (40.52%)	63 (54.31%)
Male	5 (4.31%)	48 (41.38%)	53 (45.69%)
Total	21 (18.1%)	95 (81.9%)	116 (100%)

**Table 3 children-08-00002-t003:** The distribution of oral lesions according to ethnicity.

Ethnicity/Oral Lesion	Oral Lesion Present	Oral Lesion Absent	Total
Romanian	2 (1.72%)	35 (30.17%)	37 (31.89%)
Hungarian	7 (6.03%)	27 (23.28%)	34 (29.31%)
Roma	12 (10.35%)	33 (28.45%)	45 (38.8%)
Total	21(18.1%)	95 (81.9%)	116 (100%)

**Table 4 children-08-00002-t004:** Frequency of oral self-inflicted lesions according to location.

Location of Lesion	Number	Frequency
Upper lip	6	19.36%
Lower lip	4	12.90%
Tongue	1	3.23%
Buccal mucosa	9	29.03%
Commissures	11	35.48%

## Data Availability

The data presented in this study are available on request from the corresponding author. The data are not publicly available due to privacy restrictions.

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
