# Peer review of "The Correlation between Oral Self-Harm and Ethnicity in Institutionalized Children"

_children, 2020, doi:10.3390/children8010002_

Round 1

Reviewer 1 Report

1 “The aim of the paper as presented in abstract should be related to the title (prevalence and/or correlation?)

  1. It is like the definition of OSH in the first paragraph is disconnected from the following paraph. If possible rearrange the paragraph. May better fit somewherere close to line 52
  2. Lines 42, 47, 50, 59,. Check whether the full stop mark is on its right place.
  3. 72 of our university: should be better to present the names of the University
  4. Line 73: Precise who were the legal representative whether institution manager, parents or and other legal guardians/representatives. Precise whether children had a chance to provide their own ascent to participate
  5. line 82: would be better to mention those mental disorders known to be etiologically linked to OHS
  6. The use of “we” for example at the line 69 is confusing.
  7. Is 167 the number of all children from all three institutions? There is a need to better clarify how the researchers got 116 from 167
  8. Data analysis:

            Chi-square is known for measuring overall association/correlation between two variables categorical variables. Authors should clarify whether or not they corrected for multiplicity error for follow up /post hoc comparisons. And whether the reported p-value is the adjusted one or the original p-values.

            Line 95: “ The level of statistical significance was set at a p value <0.05”. On Line 112, the p=0.07 and is said to be NOT significant.

            The asterisk on line 104 and line 107 are misleading. As those p-values have already presented in the text, it would be better to remove the asterisks in the table and the subsequent note under the tables

10. The first paragraph and paraph starting from 178 and 186 in discussion should be related to this study’s findings

11. line 204: it would be better to explain what to be unable to establish a “clear association between variables” means. Does it mean that the reported significant association in actually not?

Author Response

Dear Reviewer,

Thank you for your appreciation regarding our research and for your valuable comments on our manuscript. Bellow you may find our response to each of your observations.

Yours faithfully,

Anca Maria Pop

Response to Reviewer 1 Comments

Abstract

The aim of the paper as presented in abstract should be related to the title (prevalence and/or correlation?)

Authors’ response: We presented the aim of the paper as prevalence and correlation in order to match the title.

Introduction

It is like the definition of OSH in the first paragraph is disconnected from the following paraph. If possible rearrange the paragraph. May better fit somewherere close to line 52

Authors’ response: We removed the definition of OSH from lines 33-34 and replaced it in lines 66-67.

Lines 42, 47, 50, 59. Check whether the full stop mark is on its right place.

Authors’ response: We checked the full stop mark’s position and we think that now it is placed correctly.

Material and methods

Line 72 of our university: should be better to present the names of the University

Authors’ response: We mentioned the name of our university.

Line 73: Precise who were the legal representative whether institution manager, parents or and other legal guardians/representatives. Precise whether children had a chance to provide their own ascent to participate

Authors’ response: We mentioned who were the legal representatives of the child (institution manager/legal guardian). We mentioned that children were asked if they agreed to participate before their enrolment in the study (lines 82-83).

Line 82: would be better to mention those mental disorders known to be etiologically linked to OHS

Authors’ response: We mentioned neurologic and psychiatric disorders which may be linked to self-harm in line 92-93.

The use of “we” for example at the line 69 is confusing.

Authors’ response: We rephrased the sentence for a better understanding (line 78).

Is 167 the number of all children from all three institutions? There is a need to better clarify how the researchers got 116 from 167.

Authors’ response: 167 is the number of children from the three institutions who were assessed for eligibility. We added Figure 1 in order to explain the reduction to 116 children after applying the inclusion and exclusion criteria.

Results

Data analysis: Chi-square is known for measuring overall association/correlation between two variables categorical variables. Authors should clarify whether or not they corrected for multiplicity error for follow up /post hoc comparisons. And whether the reported p-value is the adjusted one or the original p-values.

Authors’ response: We thank the reviewer for the remark regarding the statistical analysis. We introduced the Bonferroni correction for the 3x2 contingency table (Table 3) and mentioned the original and the adjusted p values (lines 120-121).

Line 95: “The level of statistical significance was set at a p value <0.05”. On Line 112, the p=0.07 and is said to be NOT significant.

Authors’ response: We interpreted the p value=0.07 as greater than the adjusted p=0.016 and therefore considered it not significant.

The asterisk on line 104 and line 107 are misleading. As those p-values have already presented in the text, it would be better to remove the asterisks in the table and the subsequent note under the tables.

Authors’ response: We removed the asterisks and footnotes from Tables 2 and 3.

Discussion

The first paragraph and paraph starting from 178 and 186 in discussion should be related to this study’s findings.

Authors’ response: We correlated the three mentioned paragraphs with the results of our study and added lines 148-152, 194-197 and 205-207.

Line 204: it would be better to explain what to be unable to establish a “clear association between variables” means. Does it mean that the reported significant association in actually not?

Authors’ response: We declared that we were unable to establish a “clear association between variables” based on the definition of the cross-sectional design, which has several limitations, according to Donna K. Arnett, Steven A. Claas, Chapter 35 - Introduction to Epidemiology, Editor(s): David Robertson, Gordon H. Williams, Clinical and Translational Science, Academic Press, 2009, Pages 527-541, ISBN  9780123736390, https://doi.org/10.1016/B978-0-12-373639-0.00035-2: “Of the various designs available to researchers, cross-sectional studies provide the least robust evidence that a risk factor plays a causal role in disease etiology – hence the use of the word ‘association’ to cautiously describe the relationship between a risk factor and a disease.”

Reviewer 2 Report

The authors conducted a honest and straightforward cross sectional study on oral self-harm (OSH) in institutionalized children in central Romania. They characterized OSH presentations and compared OSH prevalence in 3 ethnic groups.

Authors in general should be careful with comparing ethic groups or ethnic profiling, but these authors thoroughly explained the influence of ethnic comparisons and reflected upon and referred to existing literature. 

Findings are relevant because they pay attention to severe problems in most vulnerable pediatric populations. Authors could elaborate more on the clinical implications.

Abstract: please, avoid too many synonyms and use term oral self-harm in abstract, consistent with title.

line 31: self-induced or self-inflicted (would be more appropriate than deliberate.

line 48: Do they have cognitive impairments because of institutionalization or due to other (genetic or acquired) causes? For readers from elswhere: are/have been cognitive assessments performed in clinics or institutions?

Are there confounders in studying (causal) correlation between institutionalization and OSH? Could intellectual disability be one of them?

line 101: 'Out of 116 participants, there were 53 (45.69%) boys and 63 (54.31%) girls' is a redundant sentence referring to the table.

table 1: add percentages in all cells.

line 194: re. Clinical relevance: the authors aim for early detection of emotional or early stage psychiatric problems, but what are the conseuqnces of early detection? Authors could more eloborate on possible early interventions.

Furthermore, what has been written under the clinical relevance could also be introduced in the Introduction to improve the logical structure of the manuscript and to build up the conclusion.

line 197: Would periodic oral inspection be the most appropriate clinical implication? Would information, education and empowerment be possible for these children and the people that take direct care of them? 

Strenghts and limitations: see earlier remark on confounders.

Author Response

Dear Reviewer,

Thank you for your appreciation regarding our research and for your valuable comments on our manuscript. Bellow you may find our response to each of your observations.

Yours faithfully,

Anca Maria Pop

Response to Reviewer 2 Comments

The authors conducted a honest and straightforward cross sectional study on oral self-harm (OSH) in institutionalized children in central Romania. They characterized OSH presentations and compared OSH prevalence in 3 ethnic groups.

Authors in general should be careful with comparing ethic groups or ethnic profiling, but these authors thoroughly explained the influence of ethnic comparisons and reflected upon and referred to existing literature. 

Findings are relevant because they pay attention to severe problems in most vulnerable pediatric populations. Authors could elaborate more on the clinical implications.

Abstract

Abstract: please, avoid too many synonyms and use term oral self-harm in abstract, consistent with title.

Authors’ response: We thank the reviewer for the observation. We used the term “oral self-harm” in lines 15 and 23.

Introduction

line 31: self-induced or self-inflicted (would be more appropriate than deliberate.

Authors’ response: We used “self-induced” instead of “deliberate” in line 31.

line 48: Do they have cognitive impairments because of institutionalization or due to other (genetic or acquired) causes? For readers from elswhere: are/have been cognitive assessments performed in clinics or institutions?

Authors’ response: The cognitive impairments are due to institutionalization, as the poor child-caregiver interaction negatively impacts the cognitive and social development. In our study we did not focus on cognitive impairments caused by genetic or acquired diseases. The assessments from the cited study (reference 13) were performed in care institutions.

Are there confounders in studying (causal) correlation between institutionalization and OSH? Could intellectual disability be one of them?

Authors’ response: We thank the reviewer for the observation. We introduced lines 50-53, which address possible confounding factors in studying the correlation between self-harm and institutionalization.

Results

line 101: 'Out of 116 participants, there were 53 (45.69%) boys and 63 (54.31%) girls' is a redundant sentence referring to the table.

Authors’ response: We removed the redundant sentence.

table 1: add percentages in all cells.

Authors’ response: We added percentages in Table 1.

Discussion

line 194: re. Clinical relevance: the authors aim for early detection of emotional or early stage psychiatric problems, but what are the conseuqnces of early detection? Authors could more eloborate on possible early interventions.

Authors’ response: We introduced lines 216-218 in order to emphasize the importance of early detection of NSSI.

Furthermore, what has been written under the clinical relevance could also be introduced in the Introduction to improve the logical structure of the manuscript and to build up the conclusion.

Authors’ response: We added lines 62-65 in the Introduction in order to emphasize the clinical relevance of the study.

line 197: Would periodic oral inspection be the most appropriate clinical implication? Would information, education and empowerment be possible for these children and the people that take direct care of them?

Authors’ response: Lines 221-225 were introduced in order to mention possible preventive strategies for NSSI.  

Strenghts and limitations: see earlier remark on confounders.

Authors’ response: We added lines 232-233 in which we referred to the confounding factors mentioned in Introduction.